# Intrinsic randomness in epidemic modelling beyond statistical uncertainty

Matthew J. Penn [1✉], Daniel J. Laydon [2], Joseph Penn[1], Charles Whittaker [2], Christian Morgenstern [2], Oliver Ratmann[2], Swapnil Mishra[3], Mikko S. Pakkanen[2,4], Christl A. Donnelly [1,2] & Samir Bhatt [2,3✉]

Uncertainty can be classified as either aleatoric (intrinsic randomness) or epistemic (imperfect knowledge of parameters). The majority of frameworks assessing infectious disease risk consider only epistemic uncertainty. We only ever observe a single epidemic, and therefore cannot empirically determine aleatoric uncertainty. Here, we characterise both epistemic and aleatoric uncertainty using a time-varying general branching process. Our framework explicitly decomposes aleatoric variance into mechanistic components, quantifying the contribution to uncertainty produced by each factor in the epidemic process, and how these contributions vary over time. The aleatoric variance of an outbreak is itself a renewal equation where past variance affects future variance. We find that, superspreading is not necessary for substantial uncertainty, and profound variation in outbreak size can occur even without overdispersion in the offspring distribution (i.e. the distribution of the number of secondary infections an infected person produces). Aleatoric forecasting uncertainty grows dynamically and rapidly, and so forecasting using only epistemic uncertainty is a significant underestimate. Therefore, failure to account for aleatoric uncertainty will ensure that policymakers are misled about the substantially higher true extent of potential risk. We demonstrate our method, and the extent to which potential risk is underestimated, using two historical examples.

[1] University of Oxford, Oxford, UK. [2] Imperial College London, London, UK. [3] University of Copenhagen, Copenhagen, Denmark. [4] University of Waterloo, Ontario, Canada. ✉email: matthew.penn@st-annes.ox.ac.uk; s.bhatt@imperial.ac.uk

Infectious diseases remain a major cause of human mortality. Understanding their dynamics is essential for forecasting cases, hospitalisations, and deaths, and to estimate the impact of interventions. The sequence of infection events defines a particular epidemic trajectory – the outbreak – from which we infer aggregate, population-level quantities. The mathematical link between individual events and aggregate population behaviour is key to inference and forecasting. The two most common analytical frameworks for modelling aggregate data are susceptible-infected-recovered (SIR) models[1] or renewal equation models[2,3]. Under certain specific assumptions, these frameworks are deterministic and equivalent to each other[4]. Several general stochastic analytical frameworks exist[3,5], and to ensure analytical tractability make strong simplifying assumptions (e.g. Markov or Gaussian) regarding the probabilities of individual events that lead to emergent aggregate behaviour.

We can classify uncertainty as either aleatoric (due to randomness) or epistemic (imprecise knowledge of parameters)[6]. The study of uncertainty in infectious disease modelling has a rich history in a range of disciplines, with many different facets[7–9]. These frameworks commonly propose two general mechanisms to drive the infectious process. The first is the infectiousness, which is a probability distribution for how likely an infected individual is to infect someone else. The second is the infectious period, i.e. how long a person remains infectious. The infectious period can also be used to represent isolation, where a person might still be infectious but no longer infects others and therefore is considered to have shortened their infectious period. Consider fitting a renewal equation to observed incidence data[3], where infectiousness is known but the rate of infection events $\rho(\cdot)$ must be fitted. The secondary infections produced by an infected individual will occur randomly over their infectious period $g$, depending on their infectiousness $v$. The population mean rate of infection events is given by $\rho(t)$, and we assume that this mean does not differ between individuals (although each individual has a different random draw of their number of secondary infections). In Bayesian settings, inference yields multiple posterior estimates for $\rho$, and therefore multiple incidence values. This is epistemic uncertainty: any given value of $\rho$ corresponds to a single realisation of incidence. However, each posterior estimate of $\rho$ is in fact only the mean of an underlying offspring distribution (i.e. the distribution of the number of secondary infections an infected person produces). If an epidemic governed by identical parameters were to happen again, but with different random draws of infection events, each realisation would be different, thus giving aleatoric uncertainty.

When performing inference, infectious disease models tend to consider epistemic uncertainty only due to the difficulties in performing inference with aleatoric uncertainty (e.g. individual-based models) or analytical tractability. There are many exceptions such as the susceptible-infected-recovered model, which has stochastic variants that are capable of determining aleatoric uncertainty[5] and have been used in extensive applications (e.g.[10]). However, we will show that this model can underestimate uncertainty under certain conditions. An empirical alternative is to characterise aleatoric uncertainty by the final epidemic size from multiple historical outbreaks[11,12] but these are confounded by temporal, cultural, epidemiological, and biological context, and therefore parameters vary between each outbreak. Here, following previous approaches[5], we analyse aleatoric uncertainty by studying an epidemiologically-motivated stochastic process, serving as a proxy for repeated realisations of an epidemic. Within our framework, we find that using epistemic uncertainty alone is a vast underestimate, and accounting for aleatoric uncertainty shows potential risk to be much higher. We demonstrate our method using two historical examples: firstly the 2003 severe acute respiratory syndrome (SARS) outbreak in Hong Kong, and secondly the early 2020 UK COVID-19 epidemic.

## Results

**An analytical framework for aleatoric uncertainty**. A time-varying general branching processes proceeds as follows: first, an individual is infected, and their infectious period is distributed with probability density function $g$ (with corresponding cumulative distribution function $G$). Second, while infectious, individuals randomly infect others (via a counting process with independent increments), driven by their infectiousness $v$ and a rate of infection events $\rho$. That is, an individual infected at time $l$, will, at some later time while still infectious $t$, generate secondary infections at a rate $\rho(t)v(t - l)$. $\rho(t)$ is a population-level parameter closely related to the time-varying reproduction number $R(t)$ (see Methods and[3] for further details), while $v(t - l)$ captures the individual's current infectiousness (note that $t - l$ is the time since infection). We allow multiple infection events to occur simultaneously, and assume individuals behave independently once infected, thus allowing mathematical tractability[13]. Briefly, we model an individual's secondary infections using a stochastic counting process, which gives rise to secondary infections (i.e. offspring) that are either Poisson or Negative Binomial distributed in their number, and Poisson distributed in their timing (see Supplementary Notes 3.3 and 3.4). We study the aggregate of these events (prevalence or incidence) through closed-form probability generating functions and probability mass functions. Our approach models epidemic evolution through intuitive individual-level characteristics while retaining analytical tractability. Importantly, the mean of our process follows a renewal equation[3,14,15]. Our formulation unifies mechanistic and individual-based modelling within a single analytical framework based on branching processes. Figure 1 shows a schematic of this process. Formal derivation is in Supplementary Note 3.

Randomness occurs at individual level, and there is a distribution of possible realisations of the epidemic given identical parameters. Simulating our general branching process would be cumbersome using the standard approach of Poisson thinning[16], and inference from simulation is more challenging still. Using probability generating functions, we analytically derive important quantities from the distribution of the number of infections, including the (central) moments and marginal probabilities given $\rho, g$ and $v$ (with or without epistemic uncertainty). We additionally use the probability generating function to prove general, closed-form, analytical results such as the decomposition of variance into mechanistic components, and the conditions under which overdispersion exists (i.e. where variance is greater than the mean). Finally, we derive a general probability mass function (likelihood function) for incidence.

If infection event $k = 0, \ldots, n$ occurred at time $\tau_k$ and produced $y_k$ infections, let $x_{kj}$ denote the end time of the infectious period of the $j^{th}$ infection at event $k$. Note that $\tau_0 = l$ is the time of the first infection event and $y_0 = 1$. Then the likelihood $L_{\text{InfPeriod}}$ of each infected person's infectious period is a product over all infections given by

$$L_{\text{InfPeriod}} = \prod_{k=0}^{n} \prod_{j=1}^{y_k} g(x_{kj} - \tau_k, \tau_k). \tag{1}$$

The likelihood of there being $y_k$ infections at time $\tau_k$ is given by

$$L_{\text{InfTime}} = \prod_{k=1}^{n} \left( \sum_{i=0}^{k-1} \sum_{j=1}^{y_i} \mathbb{1}_{\{x_{ij} < \tau_k\}} p_{y_k}(\tau_k, \tau_i) \right), \tag{2}$$

where $p_{y_k}(\tau_k, \tau_i)$ is the (infinitesimal) rate at which an individual infected at $\tau_i$ causes $y_k$ infections at time $\tau_k$, provided it is still

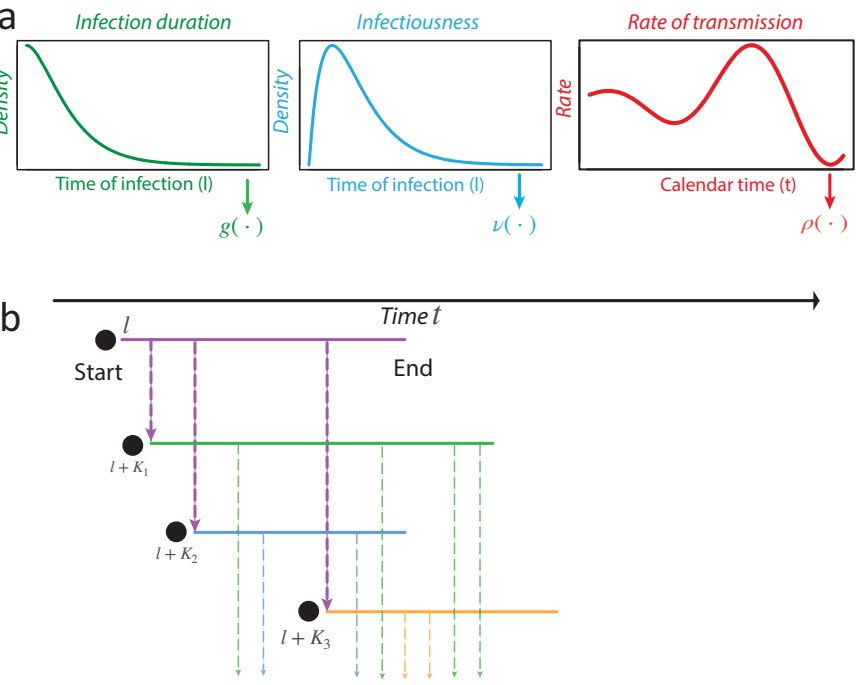

**Fig. 1 Schematic of a time-varying general branching process. a** Shows schematics for the infectious period, an individual's time-varying infectiousness (both functions of time post infection $t^*$), and the population-level mean rate of infection events. The infectious period is given by probability density function $g$. For each individual their (time-varying) infectiousness and rate of infection events are given by $\nu$ and $\rho$ respectively. In an example (**b**), an individual is infected at time $l$, and infects three people (random variables $K$, purple dashed lines) at times $l + K_1$, $l + K_2$ and $l + K_3$. The times of these infections are given by a random variable with probability density function $\sim \frac{\rho(t)\nu(t-l)}{\int_l^t \rho(u)\nu(u-l)du}$. Each new infection then has its own infectious period and secondary infections (thinner coloured lines).

infectious. Finally, the probability that no other infections occurred between the infection events at times $(\tau_k)_{k=0}^n$ is given by

$$L_{\text{Only}} = \exp\left(-\sum_{i=0}^{n}\sum_{j=1}^{y_i}\int_{\tau_i}^{\min(t,x_{ij})} r(u,\tau_i)du\right), \qquad (3)$$

where $r$ is the infection event rate and $t$ is the current time. Note the term $\exp(-x)$ comes from a Poisson assumption. Our full likelihood $L_{\text{Full}}$ is then

$$L_{\text{Full}} = L_{\text{InfPeriod}} \times L_{\text{InfTime}} \times L_{\text{Only}}. \qquad (4)$$

Full derivations of these quantities are provided in Supplementary Note 3. If discrete time is assumed, Eq. 4 simplifies to a likelihood commonly used for inference[17]. Markov Chain Monte Carlo can be used on Eq. 4 to sample aleatoric incidence realisations, but it is often simpler to solve the probability generating function with complex integration. The probability generating function, equations for the variance, and derivations of the probability mass function are found in Supplementary Notes 3, 4, 5 and 6, and a summary of the main analytical results is found in the Methods.

**The dynamics of uncertainty**. We derive the mean and variance of our branching process. The general variance Eq. 9 (see Methods) captures uncertainty in prevalence over time, where individual-level parameters govern each infection event. This equation comprises three terms: the timing of secondary infections from the infectious period (Eq. 9a); the offspring distribution (Eq. 9b); and propagation of uncertainty through the descendants of the initial individual (Eq. 9c). Importantly, this last term depends on past variance, showing that the infection process itself contributes to aleatoric variance, and does not arise only from uncertainty in individual-level events. In short, unlike common Gaussian stochastic processes, the general variance in

disease prevalence is described through a renewal equation. Therefore, future uncertainty depends on past uncertainty, and so the uncertainty around subsequent epidemic waves has memory. Additionally, uncertainty is driven by a complex interplay of time-varying factors, and not simply proportional to the mean. For example, a large first wave of infection can increase the variance of the second wave. As such, the general variance Eq. 9 disentangles and quantifies the causes of uncertainty, which remain obscured in brute-force simulation experiments[5].

Consider a toy simulated epidemic with $\rho(t) = 1.4 + \sin(0.15t)$, where the offspring distribution is Poisson in both timing and number of secondary infections, and where infectiousness $\nu$ is given by the probability density function $\nu \sim \text{Gamma}(3, 1)$, and, similarly, the infectious period $g \sim \text{Gamma}(5,1)$. Here the parameters of the Gamma distribution are the shape and scale respectively. The resulting variance is counterintuitive. We prove analytically that overdispersion emerges despite a non-overdispersed Poisson offspring distribution. The second wave has a lower mean but a higher variance than the first wave (Fig. 2), because uncertainty is propagated. If the variance were Poisson, i.e. equal to the mean, the second wave would instead have a smaller variance due to fewer infections. Initially, uncertainty from individuals is largest, but as the epidemic progresses, compounding uncertainty propagated from the past dominates [Fig. 2, bottom right]. Note that in this example with zero epistemic uncertainty (we know the parameters perfectly), aleatoric uncertainty is large.

In Eq. 9, the first two terms account for uncertainty in the infectious periods of all infected individuals. The third term denotes the uncertainty from the offspring distribution. By construction, the timing of infections is an inhomogenous Poisson process, where at each infection time the number of infections is random. The third term (Eq. 9b) contains the second

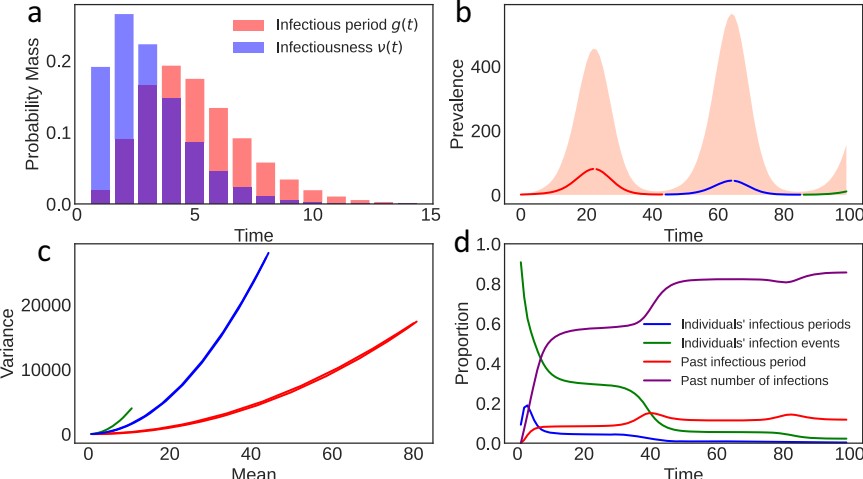

**Fig. 2 Aleatoric uncertainty without overdispersed offspring distribution.** Plots show simulated epidemic where $\rho(t) = 1.4 + \sin(0.15t)$, with a Poisson offspring distribution. We use infectiousness $\nu \sim \text{Gamma}(3, 1)$, and infectious period $g \sim \text{Gamma}(5,1)$. **a** Overlap between $g$ and the infectiousness $\nu$, where $g$ controls when the infection ends e.g. by isolation. **b** Predicted mean and 95% aleatoric uncertainty intervals for prevalence. Note there is no epistemic uncertainty as the parameters are known exactly **c** Phase plane plot showing the mean plotting against the variance. **d** Proportional contribution to the variance from the individual terms in Eq. (9). Compounding uncertainty from past events is the dominant contributor to overall uncertainty.

moment of the offspring distribution, which is the variability around its mean (i.e. $\rho(t)$). The second moment quantifies the extent of possible superspreading. In contrast to other studies[18,19], we find that individual-level overdispersion in the offspring distribution is less important than explosive epidemics. Under a null Poisson model, with no overdispersion (see Poisson case in Fig. 2), substantial aleatoric uncertainty arises from a Poisson offspring distribution combined with variance propagation. We rigorously prove via the Cauchy-Schwarz inequality that, under a mild condition on the possible spread of the epidemic, the variance of number of infections at a given time is always greater than the mean, and hence is overdispersed. Overdispersion in the offspring infection distribution is therefore not necessary for high aleatoric uncertainty, although it still increases variance at both individual-level and population-level.

We derive the conditional variance, with known past events but unknown future events. Conditional variance grows proportionally to the square of the mean, with additional terms containing the previous variance. Therefore aleatoric uncertainty grows and forecasting exercises based only on epistemic uncertainty greatly underestimates the risk of very large epidemics, and this underestimation becomes more severe as the forecast horizon expands or as the epidemic grows.

**Aleatoric uncertainty in the SARS 2003 epidemic.** To demonstrate the importance of aleatoric uncertainty, we analyse daily incidence of symptom onset in Hong Kong during the 2003 severe acute respiratory syndrome (SARS) outbreak[20–22]. The epidemic struck Hong Kong in March-May 2003, with a case fatality ratio of 15%. We fit a Bayesian renewal equation assuming a random walk prior distribution for the rate of infection events $\rho$[3], using Eq. 4 for inference. We ignore $g$ and assume that the distribution of generation times mirrors the distribution of infectiousness, i.e. that the infectiousness $\nu$ equals the generation time[20]. Note these parameter choices are illustrative and do not affect our main conclusions. The fitted $\rho(t)$ in Fig. 3 (top left) shows two major peaks, consistent with the major transmission events in the epidemic[22]. Figure 3 (top right) shows the mean epistemic fit, with epistemic (posterior) uncertainty tightly distributed around the data. Figure 3 (bottom left) shows the aleatoric uncertainty under optimistic and pessimistic scenarios (i.e.

the upper and lower bounds of $\rho(t)$ in Fig. 3 (top right)). The pessimistic scenario includes the possibility of extinction, but also an epidemic that could have been more than six times larger than that observed. The optimistic scenario suggests we would observe an epidemic of at worst comparable size to that observed. Finally, Fig. 3 (bottom right) shows epistemic and aleatoric forecasts at day 60 of the epidemic, fixing $\rho(t)$ using the 95% epistemic uncertainty interval to be constant at either $\rho(t \geq 60) = 0.38$ or $\rho(t \geq 60) = 0.83$ and simulating forwards. While the epistemic forecast does contain the true unobserved outcome of the epidemic, it underestimates true forecast uncertainty, which is 1.3 times larger. The range of the constant $\rho$ for forecast is below 1, and yet we still see substantial aleatoric uncertainty. If $\rho$ were above 1 for a sustained period, aleatoric uncertainty would play a smaller role[23], but this is rare with real epidemics, where susceptible depletion, behavioural changes or interventions keep $\rho$ around 1. Our results therefore highlight that epistemic uncertainty drastically underestimates potential epidemic risk.

**Aleatoric risk assessment in the early 2020 COVID-19 pandemic in the UK.** To demonstrate the practical application of our model, we retrospectively examine the early stage of the COVID-19 pandemic in the UK, using only information available at the time. While the date of the first locally transmitted case in the UK remains unknown (likely mid-January 2020[24]), COVID-19 community transmission was confirmed in the UK by late January 2020, and we therefore start our simulated epidemic on January 31st 2020. We consider uncertainty in the predicted number of deaths on March 16th 2020[25], during which time decisions regarding non-pharmaceutical interventions were made. Testing was extremely limited during this period, and COVID-19 death data were unreliable. For this illustration, we assume that we did not know the true number of COVID-19 deaths, as was the case for many countries in early 2020. Policymakers then needed estimates of the potential death toll, given limited knowledge of COVID-19 epidemiology and unreliable national surveillance.

We simulated an epidemic from a time-varying general branching process with a Negative Binomial offspring distribution, using parameters that were largely known by March 16th 2020 (Table 1). The infection fatality ratio, infection-to-onset

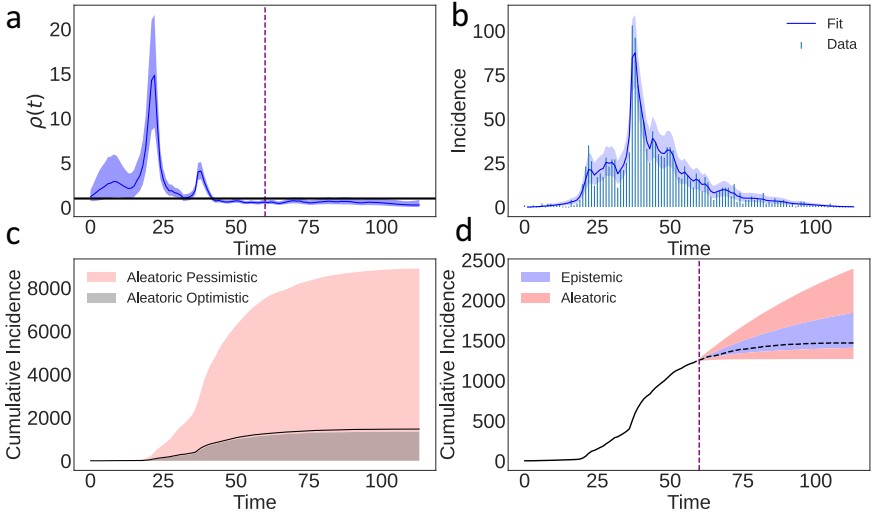

**Fig. 3 The 2003 SARS epidemic in Hong Kong[20,21]. a** $\rho(t)$ with 95% epistemic uncertainty. **b** Fitted incidence mean, 95% epistemic uncertainty with observational noise from using Eq. (4). Data is daily incidence of symptom onset. **c** Aleatoric uncertainty from the start of the epidemic under an optimistic and pessimistic $\rho(t)$. **d** Epistemic (blue) and epistemic and aleatoric uncertainty (red) while keeping $\rho$ constant at the forecast data (dotted line). Forecasting is from day 60.

| Table 1 Epidemiological parameters available on March 16th 2020 used in branching process simulation. | | |
|---|---|---|
| **Epidemiological Parameter** | **Value or Distribution** | **Citation** |
| Infection Fatality Ratio | 0.9% | 39,40 |
| Basic Reproduction Number | $2-4$ | 25,41 |
| Serial Interval Distribution | $\sim \mathrm{Gamma}\left(7.82, \frac{1}{0.62}\right)$ | 30,39,42 |
| Onset-to-Death Distribution | $\sim \mathrm{Gamma}(1.45, 10.43)$ | 39,43 |
| Infection-to-Onset Distribution | $\sim \mathrm{Gamma}(35.16, 6.9)$ | 30,39 |
| Overdispersion Coefficient | 0.53 | 44 |

distribution and onset-to-death distribution were convoluted with incidence[3] to estimate numbers of deaths. Estimated COVID-19 deaths and uncertainty estimates between January 31st and March 16th 2020 are shown in Fig. 4 (Top). While the epistemic uncertainty contains the true number of deaths, it is still an underestimate, and including aleatoric uncertainty, we find that the epidemic could have had more than four times as many deaths. Consider a hypothetical intervention on March 17th 2020 (Fig. 4 (bottom)) that completely stops transmission. Deaths would still occur from those already infected but no new infections would arise. In this hypothetical case, the aleatoric uncertainty would still be 2.5 times the actual deaths that occurred (when in fact transmission was never zero or close to it). This hypothetical scenario highlights the scale of aleatoric uncertainty, and demonstrates that our method can be useful in assessing risk in the absence of data by giving a reasonable worst case. Further, we observe that using only epistemic uncertainty provides a reasonably good fit in a relatively short time-horizon (Fig. 4, Top), but soon afterwards greatly underestimates uncertainty (Fig. 4, Bottom). The fits using aleatoric uncertainty provide a more reasonable assessment of uncertainty. While we concentrate on the upper bound, the lower bound on the worst-case scenario still exceeds zero, and therefore the epidemic going extinct by March 16th in the worst-case with no external seeding would have been very unlikely. Aleatoric uncertainty highlights a more informative reasonable worst-case estimate than epistemic uncertainty alone, and could be a useful metric for a policymaker in real time, with low-quality data, without requiring simulations from costly, individual-based models.

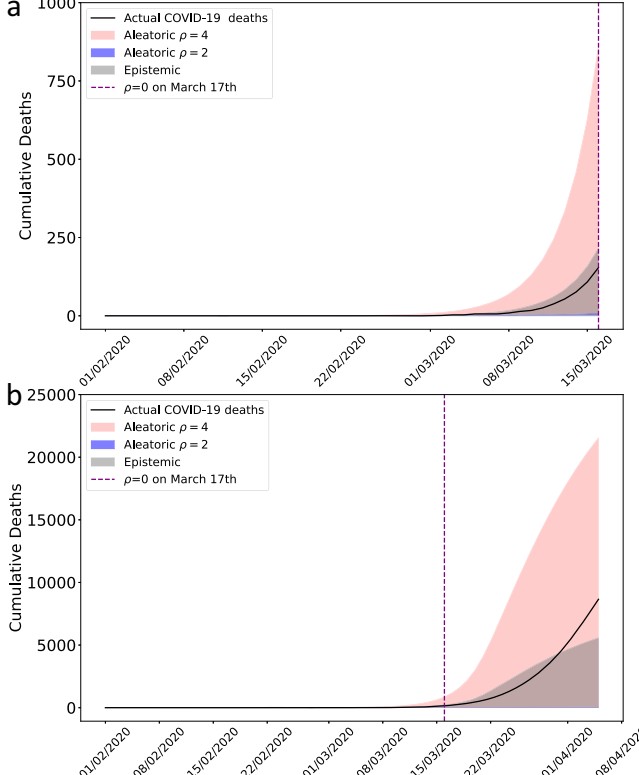

**Fig. 4 Early 2020 COVID-19 pandemic in the UK. a** shows a simulated epidemic using parameters available on March 16th 2020 (Table 1), for a plausible range of $\rho = R_0$ between 2 and 4. Blue bars indicate actual COVID-19 deaths, which we assume no knowledge of. The purple line is March 17th 2020, we set transmission to zero i.e. $\rho = 0$, to simulate an intervention that stops transmission completely. The grey envelope is the epistemic uncertainty and the red envelope the aleatoric uncertainty. **b** is the same as the top plot, except time is extended past March 17th with transmission being zero. Note aleatoric uncertainty is presented but is very close to zero.

## Discussion

Stochastic models more realistically model natural phenomena than deterministic equations[26], and particularly so with infection processes[27]. Accordingly, individual-based models have found much success[28,29] in capturing the complex dynamics that emerge from infectious disease outbreaks, and have been highly influential in policy[25]. However, despite a plethora of alternatives, many analytical frameworks still tend to be deterministic[21,30,31], and only consider statistical, epistemic parameter uncertainty. Frameworks that expand deterministic, mechanistic equations to include stochasticity use a Gaussian noise process[5], or restrict the process to be Markovian. Markovian branching processes require the infection period or generation time to be exponentially distributed - a fundamentally unrealistic choice for most infectious diseases. Further, a Gaussian noise process is unlikely to be realistic[12].

Our results show that individual-level uncertainty is overshadowed by uncertainty in the infection process itself. Profound overdispersion in infectious disease epidemics is not simply a result of overdispersion in the offspring distribution, but is fundamental and inherent to the branching process. We rigorously prove that even with a Poisson offspring distribution (not characterised by overdispersion), overdispersion in resulting prevalence or incidence is still virtually always guaranteed. We show that forecast uncertainty increases rapidly, and therefore common forecasting methods almost certainly underestimate true uncertainty. Similar to other existing frameworks, our approach provides a different methodological tool to evaluate uncertainty in the presence of little to no data, assess uncertainty in forecasting, and retrospectively assess an epidemic. Other approaches, such as agent based models, could also be readily used. However, the framework we present permits the unpicking of dynamics analytically and from first principles without a black box simulator. Equally, this is also a limitation, since new and flexible mechanisms cannot be easily integrated or considered.

We have considered only a small number of mechanisms that generate uncertainty. Cultural, behavioural and socioeconomic factors could introduce even greater randomness. Therefore our framework may underestimate true uncertainty in infectious disease epidemics. The converse is also likely, contact network patterns and spatial heterogeneity also limit the routes of transmission, such that the variability in anything but a fully connected network will be lower. Furthermore, our assumption of homogeneous mixing and spatial independence overestimates uncertainty. A sensible next step for future research to to study the dynamics of these branching processes over complex networks. Finally at the core of all branching frameworks in an assumption of independence, which is unlikely to be completely valid (people mimic other people in their behaviour) but is necessary for analytical tractability. Studying the effect of this assumption compared to agent based models would also be a useful area of future research.

We provide one approach to determining aleatoric uncertainty. Other approaches based on stochastic differential equations, Markov processes, reaction kinetics, or Hawkes processes all have their respective advantages and disadvantages. The differences in model specific aleatoric uncertainty and how close the models come to capturing the true, unknown, aleatoric uncertainty is a fundamental question moving forwards. In this paper we have provided yet another approach to characterise aleatoric uncertainty, where this approach is most useful and how it can be reconciled with existing approaches will be an interesting area of study.

## Methods

Detailed derivations of the methods can be found in the Supplementary Notes, with a high level description of the content found in Supplementary Note 1.

A time-varying general branching process proceeds as follows: first, a single individual is infected at some time $l$, and their infectious period $L$ is distributed with probability density function $g$ (and cumulative distribution function $G$). Second, during their infectious period, they randomly infect other individuals, affected by their infectiousness $v(t - l)$, and their mean number of secondary infections, which is assumed to be equal to the population-level rate of infection events $\rho(t)$. $\rho(t)$ is closely related to the time-varying reproduction number $R(t)$ (see[3] for details). The infectious period $g$ accounts for variation in individual behaviour. If people take preventative action to reduce onward infections, their reduced infection period can stop transmission despite remaining infectious. Where infectious individuals do not change their behaviour, $g$ can be ignored and individual-level transmission is controlled by infectiousness $v$ only. Each newly infected individual then proceeds independently by the same mechanism as above. Specifics can be found in Supplementary Notes 2.1–2.5.

Formally, if an individual is infected at time $s$, their number of secondary infections is given by a stochastic counting process $\{N(t,s)\}_{t \geq s}$, which is independent of other individuals and has independent increments. We assume here that the epidemic occurs in continuous time, and hence that $N(t, s)$ is continuous in probability, although we consider discrete-time epidemics in Supplementary Note 7. To aid calculation, we suppose $N(t, s)$ can be defined from a Lévy Process $\mathcal{N}(t)$—that is, a process with both independent and identically distributed increments—via $N(t, s) = \mathcal{N}(\int_s^t r(k, s)dk)$ for some non-negative rate function $r$. It is assumed that each counting process $\{N(t, s)\}_{t \geq s}$ is defined from an independent copy of $M(t)$. This formulation has two advantages: first, the dependence of $N(t, s)$ on $s$ is restricted to the rate function $r$; and second, if $J_{\mathcal{N}}(t)$ counts the number of infection events in $\mathcal{N}(t)$ (where here infection events refer to an increase, of any size, in $N(t, s)$), then $J_{\mathcal{N}}(t)$ is a Poisson process with some rate $\kappa$[32]. We can then define $J(t, s)$ to be the counting process of infection events in $N(t, s)$, and $Y(v)$ to be size of the infection event (i.e. the number of secondary infections that occur) t time $v$. We assume that $Y$ is independent of $s$, although such a dependence would curtail superspreading to depend on infectiousness, and could be incorporated into the framework. Therefore $J(t, s)$ is an inhomogeneous Poisson Process (and so $N(t, s)$ has been characterised as an inhomogeneous compound Poisson Process). We consider the cases where $N(t, s)$ is itself an inhomogeneous Poisson process, and where $N(t, s)$ is a Negative Binomial process. This allows us to examine effects of overdispersion in the number of secondary infections, although our framework allows for more complicated distributions.

Here, $r(t, l) = \rho(t)v(t - l)$ where $\rho(t)$ models the population-level rate of infection events, and $v(t - l)$ models the infectiousness of an individual infected at time $l$. If $v(t - l)$ is sufficiently well characterised by the generation time (i.e. where the timing of secondary infections mirrors tracks their infectiousness), and the infectious period can be ignored, then the integral $\int_l^t r(s, l)ds$ has the same scale as the commonly used reproduction number $R(t)$[3]. The branching process yields a series of birth and death times for each individual (i.e. the time of infection and the end of the infectious period respectively), from which prevalence (the number of infections at any given time) or cumulative incidence (the total number of infections up to any time) can be defined.

**Probability generating function**. We derive the probability generating function for a time-varying age-dependent branching process, allowing derivation of the mean and higher-order moments (full derivations can be found in Supplementary Notes 3.1–3.7). We consider two special cases for the number of new infections $Y(v)$ at each infection event: a Poisson distribution and a logarithmic (log series) distribution. In both cases, we assume that the distribution of $Y(v)$ is equal for all values of $v$. In the Poisson case, the number of new infections at each infection time is, by definition, one. Therefore the number of infections an individual creates is Poisson distributed, and closely clustered around the mean rate of infection events. The logarithmic case, which causes $N(t, l)$ to be a Negative Binomial process, more realistically allows multiple infections to occur at each infection time, and so the number of infections an individual causes is overdispersed. The pgf (probability generating function), $F(t, l; s) = E(s^{Z(t,l)})$, can be derived by conditioning on the lifetime, $L$, of the first individual. That is,

$$E\left(s^{Z(t,l)}\right) = (1 - G(t - l, l))E(s^{Z(t,l)}|L \geq t - l) + \int_0^{t-l} E(s^{Z(t,l)}|L = u)g(u, l)du$$

(5)

Note that if the individuals directly infected by the initial individual are infected at times $l + t_1, \dots, l + t_n$, then

$$Z(t, l) = 1 + \sum_{i=1}^{n} Z(t, l + t_i)$$

(6)

This observation allows us to write the generating function $F(t, l)$ as a function of $F(t, u)$ for $u \in (t, l)$. As $F(t, t) = s$, this allows us to iteratively find the value of $F(t, l)$.

Explicitly, we have

$$
\begin{aligned}
\underbrace{F(t,l;s)}_{\text{pgf}} = & \underbrace{(1-G(t-l,l))}_{P(L>t-l)} \, q_1 \Bigg( \int_0^{t-l} \underbrace{\overbrace{f(\quad \underbrace{F(t,l+k;s)}_{\text{pgf of process started at }l+k})}^{\text{pgf of }Y} \overbrace{\rho(l+k)\nu(k)dk}^{\text{infection rate at time }l+k}}_{\text{pgf of }J(t,l)} \Bigg) \\
& + \int_0^{t-l} \underbrace{q_2 \Bigg( \int_0^u \underbrace{\overbrace{f(\quad \underbrace{F(t,l+k;s)}_{\text{pgf of process started at }l+k})}^{\text{pgf of }Y} \overbrace{\rho(l+k)\nu(k)}^{\text{infection rate at time }l+k} \quad dk}_{\text{pgf of }J(l+u,l)} \Bigg) \underbrace{g(u,l)du}_{P(L=l+u)},}
\end{aligned}
\tag{7}
$$

where $q_1(z;s) = se^z$, and where $q_2(z) = e^z$ in the case where $Z(t,l)$ refers to prevalence, whereas $q_2(z;s) = se^z$ in the case where $Z(t,l)$ refers to cumulative incidence. Note also that $f(z) = z - 1$ in the Poisson case and $f(z) = -\phi \left[ \log\left(1 - (1 - \frac{\phi}{1+\phi}z)\right) - \log\left(\frac{\phi}{1+\phi}\right) \right]$ in the log-series case and that the constant $\kappa$ is absorbed into $\rho$.

The key intuition in understanding Eq. 7 is that for an integer random variable $X$ and iid (independent and identically distributed) random variables $Y_i$, $E(s^{\sum_{i=1}^X Y_i}) = G_X(G_Y(s))$, where $G_X$ and $G_Y$ are the generating functions of $X$ and $Y_i$ respectively. Thus, we expect the pgfs of the various parts of our model to combine via composition, as occurs in the equation above.

Mean incidence can be recovered from both prevalence (via back calculation[3]) and cumulative incidence. In Eq. 7 for the Negative Binomial case, $\phi$ is the degree of overdispersion. Equation (7) is solvable using via quadrature and the fast Fourier transform via a result from complex analysis[33] and scales easily to populations with millions of infected individuals, and the probability mass function can be computed to machine precision (a full derivation is available in Supplementary Note 3.7).

**Variance decomposition**. For simplicity, we only summarise the decomposition for prevalence, but an analogous and highly similar derivation for cumulative incidence can be found in Supplementary Note 3.5. We can derive an analytical equation for the mean and variance of the entire branching process (full derivations can be found in Supplementary Notes 4.1–4.7 and the mathematical properties of the variance equations can be found in Supplementary Notes 6.1–6.3). The mean prevalence $M(t,l)$ is given by

$$
M(t,l) = (1 - G(t-l,l)) + \int_0^{t-l} M(t,l+u)\rho(l+u)\nu(u)\mathbb{E}(Y)(1-G(u,l))du.
\tag{8}
$$

Note, $\rho$ can be scaled to absorb the $E(Y)$ and $\kappa$ constants. Equation (8) is consistent with that previously derived in[3]. The second moment, $W(t,l) := \mathbb{E}(Z(t,l)(Z(t,l)-1))$ allows us to determine the variance, $V(t,l)$ as $V(t,l) = W(t,l) + M(t,l) - M(t,l)^2$. The variance can be decomposed into three mechanistic components.

$$
\begin{aligned}
V(t,l) = & \underbrace{\int_0^{t-l} \left[ \int_0^u M(t,l+k)\rho(l+k)\nu(k)dk \right]^2 g(u,l)du - M(t,l)^2}_{\text{(9a): uncertainty from the infectious period}} \\
& + \underbrace{(1-G(t-l,l)) \left[ 1 + 2\int_0^{t-l} M(t,l+u)\rho(l+u)\nu(u)du + \left( \int_0^{t-l} M(t,l+u)\rho(l+u)\nu(u)du \right)^2 \right]}_{\text{(9a continued): uncertainty from the infectious period}} \\
& + \underbrace{\int_0^{t-l} M(t,l+u)^2 \mathbb{E}(Y^2)\rho(l+u)\nu(u)(1-G(u,l))du}_{\text{(9b): uncertainty from the offspring distribution}} \\
& + \underbrace{\int_0^{t-l} V(t,l+u)\rho(l+u)\nu(u)(1-G(u,l))du}_{\text{(9c): uncertainty propagated from the past}} .
\end{aligned}
\tag{9}
$$

The general variance Eq. 9 captures the evolution of uncertainty in population-level disease prevalence over time, where fixed individual-level disease transmission parameters govern each infection event. Unlike the simple Galton–Watson process, we find that previously unknown factors also determine aleatoric variation in disease prevalence. Specifically, the general variance Eq. 9 comprises three terms, one for the infectious period (Eq. 9a), one for the number and timing of secondary infections (Eq. 9b), and a term that propagates uncertainty through descendants of the initial individual (Eq. 9c). Importantly, the last term (Equation 9c) depends on past variance, showing that the infection process itself contributes to aleatoric variance, and this is distinct from the uncertainty in individual infection events. In short, and unlike Gaussian stochastic processes, the general variance in disease prevalence is described through a renewal equation. Intuitively then, uncertainty in an epidemic's future trajectory is contingent on past infections, and that the uncertainty around consecutive epidemic waves are connected. As such, the general variance Eq. 9 allows us to disentangle important aspects of infection dynamics that remain obscured in brute-force simulations[5].

**Overdispersion**. We define an epidemic to be expanded if at time $t$ there is a non-zero probability that the prevalence, not counting the initial individual or its secondary infections, is non-zero.

Note that this is a very mild condition on an epidemic - in a realistic setting, the only way for an epidemic to not be expanded is if it is definitely extinct by time $t$, or if $t$ is small enough that tertiary infections have not yet occurred.

Large aleatoric variance intrinsic to our branching process implies that the prevalence of new infections (that is, prevalence excluding the deterministic initial case) is always strictly overdispersed at time $t$, providing the epidemic is expanded at time $t$. A full proof is given in Supplementary Note 4.4, but we provide here a simpler justification in the special case that $G(t-l,l) = 1$.

In this case, prevalence of new infections is equal to standard prevalence, and the equations for $M(t,l)$ and $V(t,l)$ simplify significantly. Switching the order of integration in the equation for $M(t,l)$ gives

$$
\begin{aligned}
M(t,l) &= \int_0^{t-l} M(t,l+u)\rho(l+u)\nu(u)\mathbb{E}(Y)(1-G(u,l))du \\
&= \int_0^{t-l} \left[ \int_0^u M(t,l+k)\rho(l+k)\mathbb{E}(Y)\nu(k)dk \right] g(u,l)du
\end{aligned}
\tag{10}
$$

and hence, the Cauchy-Schwarz Inequality shows that

$$
M(t,l)^2 \leq \left( \int_0^{t-l} \left[ \int_0^u M(t,l+k)\mathbb{E}(Y)\rho(l+k)\nu(k)dk \right]^2 g(u,l)du \right)
\tag{11}
$$

as $\int_0^{t-l} g(u,l)du = 1$. Thus, the first term, (Eq. 9a), in the variance equation is non-negative.

The remaining terms can be dealt with as follows. (Eq. 9a) is equal to zero, and the sum of (Eq. 9c) is (using $Y(l+u,l)^2 \geq Y(l+u,l)$) bounded below by $\int_0^{t-l} \mathbb{E}(Z(t,l+u)^2)\mathbb{E}(Y)\rho(l+u)\nu(u)(1-G(u,l))du$. Finally, noting that $Z(t,l+u)^2 \geq Z(t,l+u)$, this is bounded below by $\int_0^{t-l} M(t,l+u)\mathbb{E}(Y)\rho(l+u)\nu(u)(1-G(u,l))du = M(t,l)$. Hence, $V(t,l) \geq M(t,l)$ holds.

To show strict overdispersion, note that for $V(t,l) = M(t,l)$ to hold, it is necessary that

$$
\begin{aligned}
& \int_0^{t-l} \mathbb{E}(Z(t,l+u)^2)\mathbb{E}(Y)\rho(l+u)\nu(u)(1-G(u,l))du \\
& = \int_0^{t-l} M(t,l+u)\mathbb{E}(Y)\rho(l+u)\nu(u)(1-G(u,l))du
\end{aligned}
\tag{12}
$$

and hence, for each $u$ (as $\mathbb{E}(Y) > 0$)

$$
\mathbb{E}[Z(t,l+u)(Z(t,l+u)-1)] = 0 \quad \text{or} \quad \rho(l+u)\nu(u)(1-G(u,l)) = 0
\tag{13}
$$

If new infections can be caused, then more than one new infection can be caused. Thus, if an individual infected at $l+u$ has $\mathbb{E}[Z(t,l+u)(Z(t,l+u)-1)] = 0$, this individual cannot cause new infections whose infection trees have non-zero prevalence at time $l+u$. Hence, the condition (13) is equivalent to the epidemic being non-expanded at time $t$, as at each time $l+u$, either no infections are possible from the initial individual, or any individuals that are infected at time $l+u$ contribute zero prevalence at time $t$ from the new infections they cause.

Hence, $Z(t,l)$ is strictly overdispersed for expanded epidemics. This means that Gaussian approximations are unlikely to be useful.

**Variance midway through an epidemic**. It is important to calculate uncertainty starting midway through an epidemic, conditional on previous events. This derivation is significantly more algebraically involved than the other work in this paper. For simplicity, we assume that $N(t,l)$ is an inhomogeneous Poisson Process, and that $L = \infty$ for each individual.

Suppose that prevalence (here equivalent to cumulative incidence) $Z(t,l) = n+1$. We create a strictly increasing sequence $l = B_0 < B_1 < \cdots < B_n$ of $n+1$ infection times, which has probability density function

$$
\underbrace{f_B(b)}_{\text{Joint pdf}} = \underbrace{\frac{1}{P(Z(t,l)=n+1)}}_{\text{normalising constant}} \underbrace{\prod_{i=1}^n \left( \rho(b_i) \sum_{j=0}^{i-1} \nu(b_i - b_j) \right)}_{\text{infection rates at each }b_i} \\
\underbrace{\exp\left[ -\sum_{i=0}^n \int_0^{t-b_i} \rho(s+l)\nu(s)ds \right]}_{\text{probability of no other infections}},
\tag{14}
$$

where pdf is short for probability mass function. Then, the variance at time $t+s$ is given by

$$
\underbrace{\text{var}(Z(t+s,l))}_{\text{variance}} = \underbrace{\int_{b=0}^t \sum_{i=0}^n V^*(t+s,b)f_{B_i}(b)db}_{\text{variance from subsequent cases}} \ldots \\
\ldots + \underbrace{\int_{b=0}^t \int_{c=0}^t \sum_{i=0}^n \sum_{j=0}^n M^*(t+s,b)M^*(t+s,c)(f_{B_i,B_j}(b,c) - f_{B_i}(b)f_{B_j}(c))dbdc,}_{\text{variance from unknown infection times}}
\tag{15}
$$

where $M^*(t+s,b)$ and $V^*(t+s,b)$ are the mean and variance of the size of the infection tree (i.e. prevalence or cumulative incidence) at time $t+s$, caused by an individual infected at $b$, ignoring all individuals they infected before time $t$.

These quantities are calculated from $M$ and $V$. Note also that $f_{B_i}$ and $f_{B_i,B_j}$ are the one-and-two-dimensional marginal distributions from $f_B$.

**Bayesian inference for SARS epidemic in Hong Kong.** The data for the SARS epidemic in Hong Kong consist of 114 daily measurements of incidence (positive integers), and an estimate of the generation time[34] obtained via the R package EpiEstim[17]. We ignore the infectious period $g$ and set the infectiousness $v$ to the generation interval. The inferential task is then to estimate a time varying function $\rho$ from these data using Eq. 4. As we note in Eq. 4 and in Supplementary Note 5 and 7.1–7.4, discretisation simplifies this task considerably. Our prior distributions are as follows

$$\phi \sim \text{Normal}^+(0, 1)$$
$$\sigma \sim \text{Exponential}(100)$$
$$\epsilon \sim \text{Normal}(0, \sigma)$$
$$\rho(t) = \rho(t-1) + \epsilon_t$$

where $\rho$ is modelled as a discrete random walk process. The renewal likelihood in Eq. 4 is vectorised using the approach described in[3]. Fitting was performed in the probabilistic programming language Numpyro, using Hamiltonian Monte Carlo[35] with 1000 warmup steps and 6000 sampling steps across two chains. The target acceptance probability was set at 0.99 with a tree depth of 15. Convergence was evaluated using the RHat statistic[36].

Forecasts were implemented through sampling using MCMC from Eq. 4. In order to use Hamiltonian Markov Chain Monte Carlo, we relax the discrete constraint on incidence and allow it to be continuous with a diffuse prior. We ran a basic sensitivity analysis using a Random Walk Metropolis with a discrete prior to ensure this relaxation was suitable. In a forecast setting, incidence up to a time point ($T = 60$) is known exactly and given as $y^{t \leq T}$, and we have access to an estimate for $\rho(t > T)$ in the future. In our case we fix $\rho(t > T) = \rho(T)$.

Our code is available at available at https://github.com/MLGlobalHealth/uncertainity_infectious_diseases.git.

**Numerically calculating the probability mass function via the probability generating function.** Following[37] and[38] (originally from[33]), the probability mass function $p$ can be recovered through a pgf $F$'s derivatives at $s = 0$. i.e. $\mathbb{P}(n) = \frac{1}{n!}\left(\frac{d}{ds}\right)^n F(s; t, \tau)|_{s=0}$ This is generally computationally intractable. A well-known result from complex analysis[33] holds that $f^{(n)}(a) = \frac{n!}{2\pi i}\oint \frac{f(z)}{(z-a)^{n+1}}\,dz$ and therefore $\mathbb{P}(n) = \frac{1}{2\pi i}\oint \frac{F(z;t,\tau)}{z^{n+1}}\,dz$ This integral can be very well approximated via trapezoidal sums as $\mathbb{P}(n) = \frac{1}{Mr^n}\sum_{m=0}^{M-1} F(re^{2\pi im/M}; t, \tau)e^{-2\pi inm/M}$ where $r = 1$[38]. The probability mass function for any time and $n$ can be determined numerically. One needs $M \geq n$, which requires solving $n$ renewal equations for the generating function and performing a fast Fourier transform. This is computationally fast, but may become slightly burdensome for epidemics with very large numbers of infected individuals (millions). A derivation of this approximation is provided in the Supplementary Note 3.7.

## Data availability
Data from Fig. 3 is available via the R-Package EpiEstim[21], and data from Fig. 4 is available at https://imperialcollegelondon.github.io/covid19local and via official UK Government reporting (https://www.ons.gov.uk/).

## code availability
All model code to reproduce Figs. 2, 3 and 4 is available at https://github.com/MLGlobalHealth/uncertainity_infectious_diseases.git.

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

## Acknowledgements

S.B., C.A.D. and D.J.L. acknowledge support from the MRC Centre for Global Infectious Disease Analysis (MR/R015600/1), jointly funded by the UK Medical Research Council (MRC) and the UK Foreign, Commonwealth & Development Office (FCDO), under the MRC/FCDO Concordat agreement, and also part of the EDCTP2 programme supported by the European Union. S.B. acknowledges support from the Novo Nordisk Foundation via The Novo Nordisk Young Investigator Award (NNF20OC0059309), which also supports S.M.. S.B. acknowledges support from the Danish National Research Foundation via a chair position. S.B. and C.M. acknowledges support from The Eric and Wendy Schmidt Fund For Strategic Innovation via the Schmidt Polymath Award (G-22-63345). S.B. acknowledges support from the National Institute for Health Research (NIHR) via the Health Protection Research Unit in Modelling and Health Economics. D.J.L. acknowledges funding from Vaccine Efficacy Evaluation for Priority Emerging Diseases (VEEPED) grant, (ref. NIHR:PR-OD-1017-20002) from the National Institute for Health Research. M.J.P. acknowledges funding from a EPSRC DTP Studentship. C.W. acknowledges support from the Wellcome Trust.

## Author contributions

S.B. and M.J.P. conceived and designed the study. S.B. performed analysis with assistance from M.J.P.. M.J.P., D.J.L. and S.B. drafted the original manuscript. M.J.P. drafted the Supplementary information with assistance from J.P.. M.J.P., D.J.L., J.P., C.W., C.M., O.R., S.M., M.S.P., C.A.D., and S.B. revised the manuscript and contributed to its scientific interpretation. S.B., M.J.P. and C.A.D. supervised the work.

## Competing interests

All authors declare no competing interests.
