## [Peer review file · Communications Physics]

Intrinsic randomness in epidemic modelling beyond statistical uncertaintyThis manuscript has been previously reviewed at another Nature Portfolio journal. This document only contains reviewer comments and rebuttal letters for versions considered at Communications Physics.

Reviewers' comments:

Reviewer #1 (Remarks to the Author):

The revised manuscript has addressed the major concerns in the original manuscript. Specific details of the assumptions and the limitations of the methods have been clarified. I still have one concern that readers may have difficulty understanding and reproducing the results. A suggestion is to cite some background references on renewal and branching processes. The two arXiv papers, references [30], [34] and paper [1] (below) can be cited at the beginning of the Supplementary Material for additional background. Some minor typographical errors: 1. References in the Supplementary Material have errors. Extra "en" in reference [3]. Missing journal in reference [4]. 2. Some equations within the text have parentheses, Equation (79) and some do not, Equation 87. 3. Furthermore, our assumption of homogeneous mixing and spatial independence overestimates of uncertainty. References [1] Champredon et al. 2018. Equivalence of the Erlang-distributed SEIR epidemic model and the renewal equation. SIAM Journal on Applied Mathematics.

Reviewer #2 (Remarks to the Author):

While the paper addresses an important topic, I share the concerns raised by previous reviewers regarding the overstated claims on the novelty and applicability of the results and the lack of detailed explanations of certain parts of the original paper.

Regarding the second concern raised by previous reviewers, I agree that the authors have made improvements to provide more detailed explanations in certain parts of the paper.

However, I still have reservations about the claims made by the authors about the extent to which the problem of aleatoric uncertainty in infectious disease models is unknown. While it is true that some research communities may focus more on epistemic uncertainty, the problem of aleatoric uncertainty is well-known and studied in other research communities. For instance, the community of physicists is particularly interested in this subject. In fact, the three references mentioned by reviewer 3 that have now been included in the paper were all written by people who are physicists or are affiliated to some extent with physics departments or institutes. Considering the intended audience of a journal such as *Communication Physics*, I believe that the tone of some statements should be lowered. Note that this of course is not just limited to the physics community, see for instance "Stochastic Epidemic Models with Inference" edited by Tom Britton and Etienne Pardoux.

I also believe that the title of the paper is somewhat overstated given that the results only apply to a specific type of model and are not directly applicable to other frameworks. In fact, even though in the abstract they state "The majority of frameworks assessing infectious disease risk consider only epidemic uncertainty" on page 14 they add several adjectives to those frameworks when they say that "analytical frameworks still tend to be deterministic, and only consider statistical, epistemic parameter uncertainty". While the section "Towards a more realistic quantification of uncertainty" provides a better overview of the problem, I feel that the language used in other parts of the paper overstates the novelty of the findings.

To give another example, by the end of the first paragraph on page 3 the authors say that "If an epidemic governed by identical parameters were to happen again, but with different random draws of infections events, each realization would be different, thus giving aleatoric uncertainty" followed by "Infectious disease models tend to consider epistemic uncertainty only, and many that readily consider aleatoric uncertainty are challenging to use in inference. A notable exception [...]". The language used may give the impression that aleatoric uncertainty is a relatively unknown concept in infectious disease modeling, but I have to disagree with that as the other reviewers did - or anyone that has

ever worked on epidemic spreading on complex networks for that matter.

Overall, I really liked the paper and I think it has the potential to make a valuable contribution to the field. I would also like to acknowledge the changes already implemented by the authors, but I believe that a fairer description of their modeling choice and its limitations on the first ~3 pages of the article is still necessary.

Reviewer #1

The revised manuscript has addressed the major concerns in the original manuscript. Specific details of the assumptions and the limitations of the methods have been clarified. I still have one concern that readers may have difficulty understanding and reproducing the results. A suggestion is to cite some background references on renewal and branching processes. The two arXiv papers, references [30], [34] and paper [1] (below) can be cited at the beginning of the Supplementary Material for additional background.

We thank the reviewer for looking at our paper a second time, and appreciate their time. Following the reviewer's suggestion we have created a new "Background literature on renewal equations" section where we cite various important sources from mathematical literature and epidemiological literature. We also cite the paper the reviewer recommends.

\section{Background literature on renewal equations}

A common approach to modelling infectious diseases is to use the renewal equation. The early theory on the properties of the renewal equation can be found here \cite{Feller1941}. Epidemiologically derived descriptions can be found here \cite{Fraser2007,Cori2013} where the renewal equation is framed in an epidemiological framework with reference to infection processes. The link between the renewal equation and the popular susceptible-infected-recovered models can be found here \cite{Champredon2018}. The basics of branching processes can be found here \cite{Harris1963-sa}. In what follows, we will arrive at a renewal equation from first principles by first starting with the probability generating function of a general branching process.

Some minor typographical errors:

1. References in the Supplementary Material have errors. Extra "en" in reference [3]. Missing journal in reference [4].
2. Some equations within the text have parentheses, Equation (79) and some do not, Equation 87.
3. Furthermore, our assumption of homogeneous mixing and spatial independence overestimates uncertainty.

Apologies, these have all been fixed.

Reviewer #2

While the paper addresses an important topic, I share the concerns raised by previous reviewers regarding the overstated claims on the novelty and applicability of the results and the lack of detailed explanations of certain parts of the original paper.

We thank the reviewer for their time and their review. We did not intend to overstate our results, but have nevertheless modified our manuscript to temper our language. We hope this will be sufficient but please let us know if further changes are required.

Regarding the second concern raised by previous reviewers, I agree that the authors have made improvements to provide more detailed explanations in certain parts of the paper.

Thank you, we really wanted to be responsive to concerns and are glad this has been sufficient.

However, I still have reservations about the claims made by the authors about the extent to which the problem of aleatoric uncertainty in infectious disease models is unknown. While it is true that some research communities may focus more on epistemic uncertainty, the problem of aleatoric uncertainty is well-known and studied in other research communities. For instance, the community of physicists is particularly interested in this subject. In fact, the three references mentioned by reviewer 3 that have now been included in the paper were all written by people who are physicists or are affiliated to some extent with physics departments or institutes. Considering the intended audience of a journal such as Communication Physics, I believe that the tone of some statements should be lowered. Note that this of course is not just limited to the physics community, see for instance "Stochastic Epidemic Models with Inference" edited by Tom Britton and Etienne Pardoux.

We understand and have now tried to further alter the language to acknowledge the large amount of varied work that has been developed on the subject of aleatoric uncertainty. In particular, following the reviewers final comment, we have reworded relevant parts in the first three pages. Please let us know if this addresses your concerns.

I also believe that the title of the paper is somewhat overstated given that the results only apply to a specific type of model and are not directly applicable to other frameworks. In fact, even though in the abstract they state "The majority of frameworks assessing infectious disease risk consider only epidemic uncertainty" on page 14 they add several adjectives to those frameworks when they say that "analytical frameworks still tend to be deterministic, and only consider statistical, epistemic parameter uncertainty". While the section "Towards a more realistic quantification of uncertainty" provides a better overview of the problem, I feel that the language used in other parts of the paper overstates the novelty of the findings.

Yes, we agree that the title is overstated. In retrospect it was a poor choice. We propose changing it to:

Beyond Statistical Uncertainty: Intrinsic Randomness in Epidemic Modelling

We believe this is a better title and hope the reviewer agrees. We are open to suggestions in any case.

To give another example, by the end of the first paragraph on page 3 the authors say that "If an epidemic governed by identical parameters were to happen again, but with different random draws of infections events, each realization would be different, thus giving aleatoric uncertainty" followed by "Infectious disease models tend to consider epistemic uncertainty only, and many that readily consider aleatoric uncertainty are challenging to use in inference. A notable exception [...]". The language used may give the impression that aleatoric uncertainty is a relatively unknown concept in infectious disease modeling, but I have to disagree with that as the other reviewers did - or anyone that has ever worked on epidemic spreading on complex networks for that matter.

We understand the reviewers' concerns. We have tried to tone down the language throughout the manuscript where we make reference to previous work (see underlined parts for new text). As noted above we have also changed the title. Examples are below.

Several major frameworks already exist to characterise both forms of uncertainty, but the majority of modelling applications assessing infectious disease risk consider only epistemic uncertainty.

Several general stochastic analytical frameworks exist \cite{Allen2017-oo,Pakkanen2021-cc}, and to ensure analytical tractability make strong simplifying assumptions (e.g. \ Markov or Gaussian) regarding the probabilities of individual events that lead to emergent aggregate behaviour.

The study of uncertainty in infectious disease modelling has a rich history in a range of disciplines, with many different facets \cite{Castro2020-vs,Neri2021-nc,scarpino2017}. These frameworks commonly propose two general mechanisms to drive the infectious process.

When performing inference, infectious disease models tend to consider epistemic uncertainty only due to the difficulties in performing inference with aleatoric uncertainty (e.g. individual-based models) or analytical tractability. There are many exceptions such as the susceptible-infected-recovered model...

Here, following previous approaches \cite{Allen2017-oo}, we analyse aleatoric uncertainty by studying an epidemiologically-motivated stochastic process,

However, despite a plethora of alternatives, many analytical frameworks still tend to be deterministic. Similar to other existing frameworks, our approach provides a different methodological tool to evaluate uncertainty

We provide one approach to determining aleatoric uncertainty. Other approaches based on stochastic differential equations, Markov processes, reaction kinetics, or Hawkes processes all have their respective advantages and disadvantages. The differences in model specific aleatoric uncertainty and how close the models come to capturing the true, unknown, aleatoric uncertainty is a fundamental question moving forwards. In this paper we have provided yet another approach to characterise aleatoric uncertainty, where this approach is most useful and how it can be reconciled with existing approaches will be an interesting area of study.

Overall, I really liked the paper and I think it has the potential to make a valuable contribution to the field. I would also like to acknowledge the changes already implemented by the authors, but I believe that a fairer description of their modeling choice and its limitations on the first ~3 pages of the article is still necessary.

We are glad the reviewer liked our paper and we have hopefully improved our descriptions in the first 3 pages. We have tried our best to reword all the relevant text to remove the implication that our approach is the only one that characterises aleatoric uncertainty. We have done this throughout the draft, including the discussion as well as the first three pages.

REVIEWERS' COMMENTS:

Reviewer #2 (Remarks to the Author):

I have carefully reviewed the paper and I am pleased to confirm that the authors have appropriately addressed all of my concerns and suggestions. Their responses and revisions have significantly improved the quality and clarity of the paper, and I believe it is now ready for publication.